# Safety and Efficacy Evaluation In Vivo of a Cationic Nucleolipid Nanosystem for the Nanodelivery of a Ruthenium(III) Complex with Superior Anticancer Bioactivity

**DOI:** 10.3390/cancers13205164

**Published:** 2021-10-14

**Authors:** Marialuisa Piccolo, Maria Grazia Ferraro, Federica Raucci, Claudia Riccardi, Anella Saviano, Irene Russo Krauss, Marco Trifuoggi, Michele Caraglia, Luigi Paduano, Daniela Montesarchio, Francesco Maione, Gabriella Misso, Rita Santamaria, Carlo Irace

**Affiliations:** 1BioChemLab, Department of Pharmacy, School of Medicine and Surgery, University of Naples “Federico II”, Via D. Montesano 49, 80131 Naples, Italy; marialuisa.piccolo@unina.it (M.P.); mariagrazia.ferraro@unina.it (M.G.F.); 2ImmunoPharmaLab, Department of Pharmacy, School of Medicine and Surgery, University of Naples “Federico II”, Via D. Montesano 49, 80131 Naples, Italy; federica.raucci@unina.it (F.R.); anella.saviano@unina.it (A.S.); francesco.maione@unina.it (F.M.); 3Department of Chemical Sciences, University of Naples “Federico II”, Via Cintia 421, 80126 Naples, Italy; claudia.riccardi@unina.it (C.R.); irene.russokrauss@unina.it (I.R.K.); marco.trifuoggi@unina.it (M.T.); luigi.paduano@unina.it (L.P.); daniela.montesarchio@unina.it (D.M.); 4CSGI—Consorzio Interuniversitario per lo Sviluppo dei Sistemi a Grande Interfase Department of Chemistry, University of Florence, Via della Lastruccia 3, 50019 Sesto Fiorentino, Italy; 5Department of Precision Medicine, University of Campania “Luigi Vanvitelli”, Via L. De Crecchio 7, 80138 Naples, Italy; michele.caraglia@unicampania.it (M.C.); gabriella.misso@unicampania.it (G.M.)

**Keywords:** anticancer ruthenium(III) complex, cationic nanosystem, in vivo preclinical models, breast cancer cells (BCC), tumour xenograft, animal biological response

## Abstract

**Simple Summary:**

The availability of selective, effective, and safe anticancer agents is a major challenge in the field of cancer research. As part of a multidisciplinary research project, in recent years our group has proposed an original class of nanomaterials for the delivery of new anticancer drugs based on ruthenium(III) complexes. In cellular models, these nanosystems have been shown to be effective in counteracting growth and proliferation of human breast cancer cells. Compared to conventional metallochemotherapeutics such as platinum-based agents whose clinical practice is associated with serious undesirable effects, ruthenium complexes share improved biochemical profiles making them more selective towards cancer cells and less cytotoxic to healthy cells. Their combination with biocompatible nanocarriers further enhances these promising features, as here showcased by our research carried out in an animal model which underscores the efficacy and safety in vivo of one of our most promising ruthenium-based nanosystems.

**Abstract:**

Selectivity and efficacy towards target cancer cells, as well as biocompatibility, are current challenges of advanced chemotherapy powering the discovery of unconventional metal-based drugs and the search for novel therapeutic approaches. Among second-generation metal-based chemotherapeutics, ruthenium complexes have demonstrated promising anticancer activity coupled to minimal toxicity profiles and peculiar biochemical features. In this context, our research group has recently focused on a bioactive Ru(III) complex—named AziRu—incorporated into a suite of ad hoc designed nucleolipid nanosystems to ensure its chemical stability and delivery. Indeed, we proved that the structure and properties of decorated nucleolipids can have a major impact on the anticancer activity of the ruthenium core. Moving in this direction, here we describe a preclinical study performed by a mouse xenograft model of human breast cancer to establish safety and efficacy in vivo of a cationic Ru(III)-based nucleolipid formulation, named HoThyRu/DOTAP, endowed with superior antiproliferative activity. The results show a remarkable reduction in tumour with no evidence of animal suffering. Blood diagnostics, as well as biochemical analysis in both acute and chronic treated animal groups, demonstrate a good tolerability profile at the therapeutic regimen, with 100% of mice survival and no indication of toxicity. In addition, ruthenium plasma concentration analysis and tissue bioaccumulation were determined via appropriate sampling and ICP-MS analysis. Overall, this study supports both the efficacy of our Ru-containing nanosystem versus a human breast cancer model and its safety in vivo through well-tolerated animal biological responses, envisaging a possible forthcoming use in clinical trials.

## 1. Introduction

The scene of transition metal-based chemotherapeutics has so far been dominated by cisplatin and its congeners [1]. Serious and more current issues have nevertheless provided a major boost to the search for unconventional metal-based drugs. Indeed, poor selectivity for cancer cells, severe systemic toxicity and increasing chemoresistance are crucial limitations which are gradually tarnishing the historical success of platinum [2,3]. Among alternative metals selected to develop new prospective anticancer drugs, ruthenium was very popular as evidenced by the recent entry into clinical trials of complexes such as NAMI-A, KP1019, NKP1339 and TLD1443 [4,5,6,7]. In this context, several reports have hitherto focused on the promising bioactivities of other types of Ru-based complexes engaged in preclinical studies [8]. In addition, some Ru-based agents have been converted into nanomaterials to exploit their full potential [9,10]. In this frame, in previous works we have designed and developed a novel class of anticancer nucleolipid-based complexes incorporating the low molecular weight ruthenium complex AziRu (Figure 1). As amphiphiles, nucleolipids can self-assemble in nanosystems able to effectively deliver the AziRu complex in cancer cells where it shows superior antiproliferative activity [11,12,13]. In fact, the structure and properties of decorated nucleolipids have a major impact on the anticancer activities of the ruthenium core. The additional co-aggregation of the nucleolipid-based Ru complexes with zwitterionic POPC or cationic DOTAP lipids—thoroughly reviewed from a chemical point of view—has ultimately allowed establishing a suite of stable and biocompatible nanosystems as potential anticancer chemotherapeutics [14,15]. Currently, some of them are under advanced preclinical investigations [16].

The AziRu bioactive complex retains a ruthenium(III) ion as metal center and is structurally related to NAMI-A but more effective than the latter in all preclinical screenings (see Figure 1) [17,18,19]. In addition to the metal redox state, it also shares some structural and functional features with the NKP1339 complex, currently the most promising clinically investigated ruthenium-based drug [20,21]. Indeed, AziRu has been proven to significantly inhibit the proliferation of human solid tumours in preclinical tests [22,23]. Bioscreen in vitro showed a good selectivity of action against different subtypes of breast cancer (BC), wherein AziRu behaves as a multi-target agent being able to reactivate distinct cell death pathways typically suppressed in oncological diseases [16]. As other metal-based drugs, AziRu can interact and produce adducts with DNA, but the structural properties of the Ru(III) complex make conceivable further interference with specific cancer-related targets, including mitochondrial proteins implicated in the regulation of dynamic cellular processes such as death or survival. From this perspective, of significance are changes in the expression of proteins belonging to the Bcl-2 family we have documented in human breast cancer cells (BCC) and which are currently the subject of further and more in-depth studies to underscore AziRu molecular targets [24,25]. Growing evidence reveal that several members of the Bcl-2 protein family can impact on cell fate decision as well by regulating a number of apoptosis-independent pathways, thereby outlining a complex network at the base of cancer cell survival and death [24,25,26]. Beyond biocompatibility ensured by a safe delivery via nucleolipid nanosystems, selectivity towards cancer cells and low toxicity profile on healthy cells are most likely related to the ruthenium oxidation state. The Ru(III) complex is indeed believed to behave as a prodrug, and its activation coupled with the emergence of more reactive species occurs selectively by reduction within the hypoxic and acidic biological microenvironment of neoplastic lesions [27]. 

Our efforts in developing Ru-based nanosystems have been hitherto supported by very promising responses achieved in preclinical models based on the use of different molecular subtypes of human BCC, including the triple negative subtypes (TNBC) [15,16]. The latter ones are responsible for very aggressive and resistant tumour phenotypes for which there are currently no effective therapeutic protocols to counteract their invasiveness [28]. Despite important therapeutic advances in recent times, BC is still the second most widespread cancer and the primary cause of cancer death in women, where the metastatic disease accounts for most of the cancer-related deaths [16,29]. Consequently, the discovery and development of new chemotherapeutic agents represents a primary need for the World Health Organization (WHO), together with the prospect for novel unconventional therapeutic protocols that significantly limit adverse effects on patients. By preclinical investigations in cellular models, we selected from our mini-library of Ru-containing nanosystems the best formulations in terms of both efficacy and safety, to be exploited for in vivo studies as the next step of preclinical validation. Among these, the cationic AziRu-based nucleolipid formulation named HoThyRu/DOTAP has been specifically designated for this study. Indeed, in the final nanoformulation the positively charged lipid DOTAP proved high performance in stabilizing and delivering the HoThyRu complex (the molecular structure of the HoThyRu nucleolipid compound incorporating the AziRu complex is shown in Figure 1) [22]. In turn, HoThyRu has demonstrated remarkable antiproliferative bioactivity in solid tumours such as BC, associated with ruthenium IC_50_ values in the low micromolar range (e.g., 12 µM in both the human endocrine-responsive epithelial-like type breast adenocarcinoma MCF-7 cells and the TNBC MDA-MB-231 cells) [15]. Cellular uptake and trafficking, anticancer activity and mechanism of action in vitro have been widely explored in the context of preclinical evaluations on BCC, uncovering biological responses including apoptosis activation and sustained autophagy induction [16].

Moving in this direction, here we describe a preclinical study performed by means of an MCF-7 xenograft tumour model to prove safety and efficacy in treating a human BC by the HoThyRu/DOTAP formulation. The MCF-7 cell line derived xenograft (CDX) model is commonly used to study therapeutic response and cell death pathways deregulation, as well as proliferation and migration [15,30]. Moreover, MCF-7 cells are largely exploited in preclinical investigations retaining features of the original mammary epithelium tumour [31,32]. In depth exploration of animal biological responses to treatment was performed to trace a first in vivo tolerability profile for this anticancer nanosystem. Accordingly, after intraperitoneal administration of HoThyRu/DOTAP formulation, both control and tumour-bearing mice were supervised by analysis of macroscopic physiological parameters together with molecular diagnostics on blood, while biological samples from organs and tissues taken after autoptic inspection were subjected to ICP-MS experiments to evaluate ruthenium bioaccumulation. Separately, tumour lesions were carefully analysed to assess the impact of anticancer therapy. The herein presented results demonstrate an important regression of tumours following HoThyRu/DOTAP administration, in consort with animal biological responses highlighting a good tolerability profile in the benefit-risk assessment process. 

## 2. Materials & Methods

### 2.1. Preparation of the HoThyRu Complex and HoThyRu/DOTAP Nanoformulation

The here investigated ruthenium(III) complex, named HoThyRu (Figure 1d), was synthesized following previously described procedures with minor modifications [19,22,33]. Liposomes were prepared through the thin film protocol by dissolving known weighed quantities of 1,2-dioleyl-3-trimethylammoniumpropane chloride (DOTAP) and HoThyRu in chloroform and then mixing them in the desired DOTAP: HoThyRu 70:30 molar ratio. The resulting solutions were transferred in a round-bottom glass tube and the solvent was evaporated with anhydrous nitrogen to obtain a homogeneous thin film. Samples were dried under vacuum for at least 24 h to ensure the complete chloroform removal before rehydration with phosphate buffered saline (PBS, pH 7.4, Sigma, Milan, Italy), previously filtered through 0.22 μm filters, to obtain a total lipid concentration of 1 mM. Finally, samples were vortexed, briefly sonicated and extruded through polycarbonate membranes with 100 nm sized pores at least 15 times to obtain monodisperse liposome dispersion, as assessed by dynamic light scattering (DLS) control measurements (see Figure A1 for a physico-chemical characterization of the HoThyRu/DOTAP nanoformulation). Bare DOTAP liposomes were prepared as previously described [22]. 

### 2.2. Cell Cultures

Epithelial-like type human breast adenocarcinoma cells MCF-7 (belonging to the luminal A molecular subtype of breast cancers, ER+, progesterone receptor positive and HER2 negative) were purchased from ATCC (Manassas, VA, USA) and grown in DMEM (Invitrogen, Paisley, UK) supplemented with 10% fetal bovine serum (FBS, Cambrex, Verviers, Belgium), L-glutamine (2 mM, Sigma), penicillin (100 units/mL, Sigma) and streptomycin (100 μg/mL, Sigma), and cultured in a humidified 5% carbon dioxide atmosphere at 37 °C. In vitro data concerning biological effects of the HoThyRu/DOTAP formulation, HoThyRu complex and DOTAP liposome in MCF-7 cells are reported in Figure A2. Cell death pathways activation in MCF-7 by treatment with the HoThyRu/DOTAP nanoformulation is supported in Figure A3. 

### 2.3. Animals and Experimental Design

4-week-old female athymic nude Foxn1nu mice (23–26 g) were purchased from Envigo RMS (Udine, Italy) and kept in an animal care facility at a controlled temperature range between 22 ± 3 °C, humidity (50 ± 20%) and on a 12:12 h light-dark cycle (lights on at 07:00 h). All mice were acclimatized to the environmental conditions for at least 5 days before starting the xenograft experiments. They were housed in Plexiglass cages (5 mice/cage) equipped with air lids, kept in laminar airflow hoods, and maintained under pathogen-limiting conditions. Animals were maintained with free access to sterile food and water. Sterile food was purchased from Envigo (Teklad global 18% protein #2018SX, Envigo, Madison, WI, USA). Cages and water were autoclaved before use. Mice were randomly divided into seven groups (control, xenotransplanted, non-xenotransplanted treated with HoThyRu, non-xenotransplanted treated with HoThyRu/DOTAP, xenotransplanted treated with DOTAP liposome, xenotransplanted treated with HoThyRu, and xenotransplanted treated with HoThyRu/DOTAP), and then used to set up xenograft models (five or ten animals for each experimental group) and bioaccumulation and toxicity studies (five animals for each experimental group). Animal studies were conducted in accordance with the guidelines and policies of the European Communities Council and were approved by the Italian Ministry of Health (n.354/2015-PR). Protocols and procedures for in vivo studies were performed under the supervision of veterinary experts according to European Legislation. All procedures were carried out to minimize the number of animals used and their suffering.

### 2.4. Generation of Human BCC-Derived Xenograft Models in Nude Mice

At 80% confluence, MCF-7 cells were trypsinized and harvested. Cell number was determined by TC20 automated cell counter (Bio-Rad, Milan, Italy) with a specific dye (trypan blue) exclusion assay [14]. Aliquots containing 5 × 10^6^ cells were opportunely 1:3 mixed in Matrigel^®^ Matrix (Growth Factor Reduced, Corning, Bedford, MA, USA) and tumours were established by subcutaneous (*s.c.*) injection into the right flank of each mouse. Mice were randomly assigned to each of the two xenotransplanted experimental groups. 

### 2.5. Treatments In Vivo: Experimental Protocols and Therapeutic Scheme

Ruthenium treatment in vivo started two weeks post tumour implant by intraperitoneal (*i.p.*) injection, according to a standardized and tested protocol [15]. In brief, 15 mg/kg of HoThyRu/DOTAP, or of an equivalent amount in ruthenium of HoThyRu (4.5 mg/kg), contained in 300 μL of sterile water (Molecular Biology Grade Water, Corning), were administered to mice once a week for 28 days (4 weeks). In addition, other animal groups were treated with an equal volume of sterile PBS or 15 mg/kg of DOTAP liposome. After 28 days of treatments, animals were sacrificed, and tumours and organs were first appropriately collected, and then carefully weighed and photographed. All experimental procedures were carried out in compliance with the international, and national law and policies (EU Directive 2010/63/EU for animal experiments, ARRIVE guidelines and the Basel declaration including the 3R concept) and approved by the Italian Ministry of Health (n.354/2015-PR).

### 2.6. Tumour Volume Determination by Caliper Measurements

Starting a week later implantation of human BCC in nude mice (measurable subcutaneous tumours of about 350–500 mm^3^), tumour volumes in xenotransplanted mice were determined throughout the study by using an external caliper. Specifically, the largest longitudinal (length) and transverse (width) diameters were monitored and recorded every two days. Tumour volumes measurements were then calculated by the formula V = (Length × Width^2^)/2.

### 2.7. Animal Supervisions and Monitoring throughout the Preclinical Study

Animals were checked daily by the veterinarian and their state of health monitored continuously. Mice body weights were recorded every two days by MS-Analytical and Precision Balance (Mettler Toledo, Columbus, OH, USA). For xenotransplanted animals, special attention was given to the tumour size as well as to the skin area near the tumour lesion to avoid animal pain. 

### 2.8. Surgical Procedures, Harvest of Tumours and Biological Samples Collection

At the end of the study, mice were sacrificed in a chamber containing CO_2_ according to AVMA guidelines for the euthanasia of animals. Every effort was made to minimize animal pain and discomfort. Tumours, organs, and tissues (blood, heart, liver, kidneys, brain, spleen, and lungs) were meticulously collected by surgical procedures under strictly aseptic conditions following sacrifices at 4, 24, 48 h and 1-week post intraperitoneal injection of HoThyRu/DOTAP (*n* = 5 or 10 animals *per* time point). The same biological samples were also collected after 28 days of treatment (once a week) with HoThyRu or with HoThyRu/DOTAP (*n* = 5 or 10 animals per time point). All animal experiments were conducted according to the guidelines of the Institutional Animal Care and Use Committee (IACUC). After macroscopic evaluations, biological samples were catalogued and properly cryopreserved at −80 °C until analysis.

### 2.9. Ruthenium Bioaccumulation In Vivo by Inductively Coupled Mass Spectrometry (ICP-MS)

ICP-MS spectrometry was used for a highly sensitive determination of ruthenium concentrations in blood and tissues after treatment in vivo with the HoThyRu complex or with the HoThyRu/DOTAP nanosystem. Biological samples were subjected to oxidative acid digestion with a mixture of 69% nitric acid and 30% *v/v* hydrogen peroxide in 8:1 ratio, using high temperature and pressure, under a microwave assisted process. A proper dilution was made, and the suspension obtained for each sample was introduced to the plasma. The mineralized samples were recovered with ultrapure water and filtered using 0.45 μm filters. The determination of ruthenium was carried out on an Aurora M90 inductively coupled plasma mass spectrometry (ICP-MS) instrument (Bruker Daltonics, Bremen, Germany). The quantitative analysis was performed using the external calibration curve method. In the analyzed samples, the ruthenium content is expressed both as percentage of the total ruthenium administered in vivo and in absolute quantity expressed as µg/kg of body weight [15,16]. 

### 2.10. Blood Samples and Assessment of Biochemical and Hematological Parameters

Standard laboratory procedures were used for blood sampling and measurements [34]. Hematological investigations including complete blood count (CBC) test and leukocyte formula, liver and kidney toxicity test were performed on citrated and non-anticoagulated blood samples, obtained by intracardiac puncture after 4, 24, and 48 h and 7 days from a single HoThyRu/DOTAP administration (15 mg/kg, *i.p.*), as well as after repeated weekly administrations of HoThyRu/DOTAP (15 mg/kg, *i.p.*, once a week for 4 weeks). Hematological investigations were performed by CELL-DYN Sapphire (Abbott SRL, Milan, Italy). All procedures were conducted under strictly aseptic conditions. Creatinine, alanine transaminase, aspartate transaminase, total bilirubin, and azotaemia of serum samples were analyzed using a Roche COBAS C8000 Automatic Analyzer (Roche Diagnostics S.p.A., Monza, Italy) with the appropriate kits.

### 2.11. Statistical Data Analysis

Results and statistical analysis comply with the international recommendations on experimental design and analysis in pharmacology, and data sharing and presentation in preclinical pharmacology [35,36,37]. All data were presented as mean values ± SEM. Statistical analysis was performed by using one-way, or two-way ANOVA followed by Dunnett’s or Bonferroni’s for multiple comparisons. GraphPad Prism 8.0 software (GraphPad Software, San Diego, CA, USA) was used for analysis. Differences between means were considered statistically significant when *p* ≤ 0.05 was achieved. Sample size was chosen to ensure alpha 0.05 and power 0.8. Animal weight was used for randomization and group allocation to reduce unwanted sources of variations by data normalization. No animals and related ex vivo samples were excluded from the analysis. In vivo and in vitro studies were carried out to generate groups of equal size, using randomization and blinded analysis.

## 3. Results

### 3.1. In Vivo Administration of HoThyRu/DOTAP Nanosystem Inhibits Tumour Growth and Proliferation

Human BCC-derived tumour xenografts in nude mice were set up by adenocarcinoma MCF-7 cells. For in vivo treatments with HoThyRu/DOTAP formulation, the animals were enrolled 2 weeks after the injection of the cells and, in any case, not before having checked the actual development of the tumour mass, as described in the experimental section. The therapeutic scheme, reported in Figure 2a, has been conceived for intraperitoneal (*i.p.*) administration of HoThyRu/DOTAP at the dose of 15 mg/kg, once a week for 28 days. Following the same experimental protocol, in vivo treatments with the bare DOTAP liposome (DOTAP, 15 mg/kg) and the HoThyRu nucleolipid complex (HoThyRu, 4.5 mg/kg) were also performed. At the end of the study (5 weeks from the start of treatments), the survival of tumour-bearing mice was 100% for both the control group (PBS) and the treated groups (DOTAP, HoThyRu, and HoThyRu/DOTAP) (Figure 2b). Moreover, no alteration in body weights was recorded neither in single groups nor by comparison between groups (Figure 2c), and no macroscopic signs of toxicity were observed. This evidence endorses the study and, as shown in detail below, suggests HoThyRu/DOTAP as well tolerated in vivo. Even the individual components of the final nanoformulation (the HoThyRu complex and the DOTAP liposome) do not appear to have negative effects on animal health. At the experimental end point, animals were sacrificed, and the tumours appropriately collected to be analysed. In Figure 2d the explanted tumour masses from the control (PBS) and treated animal groups (DOTAP, HoThyRu, HoThyRu/DOTAP) are shown. As clearly visible, only the treatment with the HoThyRu/DOTAP nanoformulation considerably reduces the tumour mass. Treatments with the bare DOTAP liposome and not co-aggregated HoThyRu complex do not produce biological effects on tumours. In support, in vivo photographs at the beginning and the end of the treatments highlight that tumour cells proliferation in mice was inhibited exclusively by HoThyRu/DOTAP treatment with respect to the untreated group (PBS), where tumours developed considerably (Figure 2e). Indeed, differences in the weight of the explanted tumour masses between the different animal experimental groups were very evident (Figure 2f). Accordingly, starting from the second week of in vivo administrations throughout the study, tumour volumes related to the HoThyRu/DOTAP treated mice were reduced significantly compared to the other animal groups (Figure 2g). In addition to in vivo results, experimental data on the in vitro biological effects of the DOTAP liposome, the HoThyRu ruthenium nucleolipid complex and the final HoThyRu/DOTAP nanoformulation in MCF-7 cells are available in Figure A2. Consistently, fluorescence bioscreens supporting the induction of both apoptotic and autophagic cell death pathways in MCF-7 cells after incubation with the HoThyRu/DOTAP nanosystem have been supplied in Figure A3.

### 3.2. Ruthenium Plasmatic Levels Following In Vivo Treatments

To evaluate ruthenium plasmatic levels over time throughout the in vivo study, blood samples from nude mice were subjected to ICP-MS analysis according to the procedures described in the experimental section. Blood samples were prepared by intracardiac puncture after 4, 24, and 48 h, as well as after 7 days from a single HoThyRu (4.5 mg/kg, *i.p.*) or HoThyRu/DOTAP administration (15 mg/kg, *i.p.*)(Figure 3a). In addition, to appraise a prospective ruthenium accumulation in plasma following chronic administration, blood samples were also prepared after repeated weekly administrations (4.5 mg/kg of HoThyRu, *i.p.*, once a week for 4 weeks; 15 mg/kg of HoThyRu/DOTAP, *i.p.*, once a week for 4 weeks). The results showed that, after a maximum value reached at 4 h (around 30 mg/L), the ruthenium blood concentrations decreased linearly as a function of time after a single HoThyRu/DOTAP dose, while remaining detectable 7 days after administration (Figure 3, white bars). Ruthenium plasma concentrations after HoThyRu/DOTAP administration are constantly higher than those measured under the same experimental conditions following a single dose of the not co-aggregated HoThyRu complex (Figure 3b, ribbed bars). The last data plotted in the bar graph of Figure 3b (28 d, 4 doses) are referred to ruthenium plasma concentrations measured after four doses (once a week) of HoThyRu and HoThyRu/DOTAP. In this case, a significant ruthenium accumulation in blood tissue was observed, specifically following repeated doses of HoThyRu/DOTAP. These findings suggest the adopted therapeutic strategy as capable of determining reasonable HoThyRu/DOTAP concentrations in the blood stream, thus allowing a wide systemic distribution of ruthenium with an effective antiproliferative action against cancer cells.

### 3.3. Ruthenium Bioaccumulation in Mice Bearing BBC Xenograft

To uncover the fate of the ruthenium-based drug and its body sites of bioaccumulation after systemic administration of the HoThyRu/DOTAP nanosystem, as well as its ability to target tumour in vivo, we evaluated ruthenium amounts in several body districts, including tumour lesions, at the endpoint of the study. We also checked for differences in the in vivo ruthenium distribution after repeated administration of the not co-aggregated HoThyRu complex and the HoThyRu/DOTAP nanoformulation. Hence, after weekly administrations of HoThyRu (4.5 mg/kg, *i.p.*, once a week for 4 weeks) or HoThyRu/DOTAP (15 mg/kg, *i.p.*, once a week for 4 weeks), mice were sacrificed, and organs and tissues appropriately collected to analyse the ruthenium content by ICP-MS analysis. Interestingly, Figure 4a,b show that a considerably higher ruthenium quantity was found within the tumour lesions after HoThyRu/DOTAP treatment in vivo than those measured after HoThyRu treatment (15.5 ± 2% vs. 4.3 ± 1% of all the ruthenium found in the analysed data). Besides reaching tumour lesions in larger quantities, overall the nanoformulation exhibits a wider distribution in the body, whereas the not co-aggregated complex accumulates mainly in the spleen (55.5 ± 4.3%) and liver (28.3 ± 3%). Indeed, the physiologic wide perfusion of several districts coupled to the nanoformulation stability allowed large amounts of ruthenium to reach organs and tissues (Figure 4b), i.e., spleen (52 ± 3%), liver (20 ± 1.5%), and kidneys (8 ± 1%). Minor ruthenium amounts were detected in heart (1.2 ± 0.3%) and lungs (3.3 ± 0.8%) after HoThyRu/DOTAP treatment. No trace of the metal was found in brain. Once assessed the best performances of the nanoformulation in vivo with respect to the not co-aggregated HoThyRu complex, we lastly performed the same experiments to evaluate ruthenium accumulation in mice over time after a single administration of HoThyRu/DOTAP (15 mg/kg, *i.p.*). Data illustrated in Figure 4c,d are reported both as percentage of ruthenium compared to the total metal detected in the various body districts and in absolute quantity expressed as mg/kg of body weight. Approximately, at different times after HoThyRu/DOTAP administration, the metal tracking appears very similar to that observed at the endpoint of the study. Nevertheless, a significant increase of the ruthenium amount percentage was detected in kidneys at 4 h after administration. This finding is probably due to the early high plasma concentrations at short times after intraperitoneal administration which, coupled to the liposome surface charge, can have an impact on renal excretion. This would also explain why the total metal amounts detected at the systemic level after 4 and 24 h are lower than in longer times (e.g., 1 week). It should also be noted that the slight ruthenium lung bioaccumulation is detectable only after 4 weeks of treatment.

### 3.4. Blood Diagnostics and Animal Response to HoThyRu/DOTAP Administration

As shown previously, all animals treated with the HoThyRu/DOTAP nanosystem reached the endpoints of the study without apparent health complications and showed a significant remission of tumour lesions. Body weight maintenance during the trial and absence of significant signs of toxicity and/or abnormal behaviours, led to consider HoThyRu/DOTAP as well tolerated at the therapeutic regimen. In line, the autopsy findings performed at the end of the study did not show any significant biological alteration suggesting toxicity on organs and tissues (Figure 5). To investigate this aspect more in detail, haematological analyses were performed. Blood samples were appropriately collected by an intracardiac puncture after 4, 24, and 48 h, and after 7 days from a single HoThyRu/DOTAP dose (15 mg/kg, *i.p.*) to study acute toxicity. In parallel, blood samples prepared after weekly administrations of HoThyRu/DOTAP (15 mg/kg, *i.p.*, once a week for 4 weeks) were used to evaluate chronic toxicity. Blood chemical analyses of the control group (non-xenotransplanted mice) were used as reference values for a proper assessment of the serum levels of each parameter throughout the study. In this way several biochemical and haematological parameters were analysed as biomarkers of liver, kidneys, spleen, and blood function (Figure 6). Substantially, clinical analyses did not reveal alterations of the physiological conditions in the haematic framework. Treated animals showed no significant change in the levels of azotaemia and liver enzymes such as alanine aminotransferase (ALT) and aspartate aminotransferase (AST) with respect to control animals. Increased values in some parameters (i.e., creatinine and bilirubin), detectable only few days (24 and 48 h) after the administration of HoThyRu/DOTAP, rapidly re-entered in a physiological range without diverging significantly from those measured in control animals. Interestingly, at the end of treatment by repeated weekly administrations, all values taken into consideration were similar or very close to the reference ones (see also Figure A4 for clinical chemistry and haematology provided by the supplier with reference to athymic nude mice). In addition, a complete blood count (CBC) test with formula was performed (Figure 7). As before, clinical data showed no important variations between the control and the treated mice groups at different time points post administration. However, an increase in haematocrit and in total white blood cells (throughout the whole study and at the endpoint of the investigation) was observed. Nevertheless, the leukocyte formula (Figure 8) did not reveal noteworthy alterations, neither after single administrations nor after a one-month dosage regimen. Overall, these data suggest HoThyRu/DOTAP nanosystem as well tolerated in vivo, underscoring the safety of the selected therapeutic protocol.

## 4. Discussion

A comprehensive in vitro research by targeted bioscreens has powered the development first and then the selection of original Ru-based nanosystems for cancer treatment [14,15,16]. Next to preclinical research in cellular models, we have now tested one of our most promising Ru-based nanosystems in an animal model. This study thereby embodies the starting point to further broaden the usage of nucleolipid nanosystems linked to a ruthenium(III) complex, named HoThyRu, as a prospective therapeutic option for human solid tumours. Among the suite of Ru-based formulations we have hitherto developed, the cationic lipid DOTAP has been selected as particularly effective in delivering the HoThyRu complex in a stable and safe formulation, which in turn has shown very promising antiproliferative activity towards solid tumours such as BC [14,15,22]. Chemically, the nucleolipidic Ru(III) complex HoThyRu includes in its structure the low molecular weight complex AziRu, inspired to NAMI-A and other analogues, such as the clinically investigated NKP1339 [38]. The latter is currently believed as a first-in-class ruthenium-based anticancer agent against solid cancer [6,39]. NKP1339 is a pro-drug which is activated in the reductive tumour microenvironment inducing cell cycle arrest, DNA synthesis inhibition, and apoptosis via the mitochondrial pathway [20,21]. In the same way, HoThyRu behaves as a multitarget agent acting at the level of nuclei and mitochondria. We have in fact highlighted strong antiproliferative effects in vitro against various molecular subtypes of BCC [14,15,16,40]. Consequently, this study was carried out by the generation of a preclinical human BCC-derived xenograft model in athymic nude mice. MCF-7 cells, as one of the most widely used models in vitro of BC, were injected subcutaneously into the flanks of animals for tumour induction in vivo [15,30,31,32].

Experimentally, we observed excellent responses to the treatment with HoThyRu/DOTAP in the different animal groups enrolled for this study. All the animals reached the end of the study and remarkably no one showed treatment-induced toxicity. Indeed, no signs of suffering, abnormal behaviours or acute toxicity were detected during and after the therapeutic regimen. Even after repeated administrations (once a week for 4 weeks) we did not detect toxicity and animal suffering. In support of these findings, untreated xenograft-bearing mice (PBS control group) quickly developed a consistent tumour mass in conjunction with local inflammation and skin ulcerations at the lesions, thereby showing markers of veterinary distress. Under the same experimental conditions, no biological effect was found on tumours in xenograft-bearing mice after treatment with DOTAP liposomes, as well as with the nucleolipid complex HoThyRu. Conversely, treatment with the HoThyRu/DOTAP nanosystem improved the health status of xenotransplanted mice by safely contrasting tumour growth, which was considerably reduced compared to the other experimental animal groups. Once explanted, tumours from HoThyRu/DOTAP-treated animals’ group were in fact significantly reduced on macroscopic observation, as thereafter strengthened by evaluation of their weights and volumes. This evidence is very encouraging since many metal complexes including some novel Ru-based compounds have been reported to be promising and potent in vitro anticancer agents, but very few have demonstrated efficacy in in vivo models [41]. Together with an acceptable tolerability profile, the evidence of their effectiveness in animal models is a key requirement to entry clinical stages. In addition to NAMI-A and KP1019 which reached clinic in the past, and NKP-1339 which is now attracting clinical interest for its in vivo efficacy and very limited side effects, many other ruthenium-based compounds have been established as anticancer drugs [8,16,21,38]. Among these, some have demonstrated efficacy in animal models, proving to be ready, or nearly ready, for clinical trials (e.g., RAPTA family and ruthenium polypyridyl complexes) [41,42,43,44]. Moreover, these candidate drugs and their analogues have shown in vivo to preferentially accumulate in the tumour, which could potentially account for their good tolerability profile [45]. As well as a range of bioactivities, they were capable of interfering with the regulation of Bcl-2 family proteins and to reactivate the intrinsic mitochondria-mediated apoptosis, reducing tumour growth in cancer xenograft models with a systemic toxicity much lower than cisplatin [46].

The in vivo efficacy of HoThyRu/DOTAP can be correlated both to ability to reach the tumour in adequate amounts after systemic administration and to selective activation of cell death pathways counteracting cancer cell proliferation. Lodged in biomimetic liposomes, the HoThyRu complex enters massively in tumour cells where it has been shown to simultaneously trigger both intrinsic apoptosis and autophagy [14,15]. It is widely accepted that the main drawback of this and other ruthenium-based antineoplastic agents is their limited stability in biological fluids. As we have beforehand demonstrated, this process also occurs for both AziRu and its nucleolipidic derivatives including the HoThyRu complex, whose effectiveness is significantly impaired over time under physiological conditions [22]. Nevertheless, exploiting the intrinsic negative charge of the HoThyRu complex, its favourable insertion into liposomes formed by the cationic lipid DOTAP leads to a substantial prevention of the degradation kinetics observed for the low molecular weight ruthenium complexes [22,40]. Cationic liposomes can in fact form stable complexes with negatively charged therapeutic agents [47]. Additionally, they can interact with negatively charged cell surfaces more readily than classical liposomes [48]. In tumour phenotypes after neoplastic transformation, negatively charged components at the outer plasma membrane, e.g., phosphatidylserines, proteoglycans, and glycoproteins, are more abundant than in healthy cells. Therefore, cationic liposomes are endowed with superior selectivity for cancer cells than neutral or anionic liposomes [49]. Indeed, it has been demonstrated that intracellular uptake of cationic liposomes by tumour cells can be 14-fold higher than that of normal liposomes [50,51]. Accordingly, by HoThyRu/DOTAP fusion with cell membranes, the ruthenium complex can directly cross the plasma membranes and enter cancer cells in large quantity [16]. In this context, data emerging from ruthenium bioaccumulation analysis in different mice body areas are very attractive and substantially in line with evidence reported for other types of cationic nanocarriers. In addition to a predictable bioaccumulation in some organs, i.e., spleen and liver, significant amounts of ruthenium have in fact been found in tumour lesions after HoThyRu/DOTAP administration. This feature is shared in preclinical testing with other ruthenium-based drug candidates and is indicative of favourable therapeutic perspectives [41]. Selective nanodelivery by a biocompatible cationic nanosystem naturally accumulating in tumour cells further heightens these features which have been by now applied in clinic to formulate anticancer agents for selective targeting, improving the safety of anticancer drugs and considerably limiting their potential side effects [48,52]. Furthermore, local associated inflammation could play a role in HoThyRu/DOTAP accumulation in tumour microenvironment. In fact, it has been reported that physicochemical properties and surface characteristics of nanocarriers can impact on their accumulation in both tumour and inflammation sites [53,54]. In compliance, ruthenium amounts we have found in tumour explants (about 15% of the total) after a 4-week therapy are quite significant, especially when related to tumours weights with respect to the whole mouse mass. Of relevance in this framework is the comparison with outcomes from the group of xenograft-bearing mice treated with the not co-aggregated HoThyRu complex, wherein the total amount of ruthenium found in tumour lesions is significantly lower (around 4%). More generally, the ruthenium distribution in vivo after administration of the not co-aggregated HoThyRu complex appears to be considerably limited, probably by its lower stability in the biological environment.

Concerning plasma concentrations after in vivo treatments, intraperitoneal administration of HoThyRu/DOTAP allows reaching significant drug blood levels. After repeated weekly administrations, plasma concentrations become stable on values around 15 mg/L, allowing in principle for a significant drug biodistribution in the whole organism. However, we found high ruthenium plasma concentrations even after single administrations. In a chemotherapeutic perspective this can be beneficial to target both tumour cells and metastatic lesions. Once more, the substantial difference between the ruthenium levels observed in biological samples from different experimental groups of animals should be underlined. The HoThyRu/DOTAP nanoformulation is much more performing than just the HoThyRu complex in providing high plasma concentrations, both after repeated administrations and after single dose. As well as bioaccumulation in tumours, an additional contribution to the reduction of side effects may arise from consideration that ruthenium(III) complexes behave as prodrugs, requiring a sort of in vivo activation by selective reduction of the metal center [22,27]. Tumour microenvironments, characterized by hypoxic conditions and a slightly more acidic pH than healthy cells, can facilitate such a redox-dependent mechanism of activation in situ [55]. Interestingly, after a 28-d therapy no ruthenium amount was detected in the brain. Based on experimental evidence, HoThyRu/DOTAP does not cross the blood brain barrier at the used dosage regimen, or at least not detectably to cause significant ruthenium bioaccumulation in the central nervous system (CNS). Differently from platinum-based chemotherapy, which is typically associated with important neurotoxicity, disposal of novel antineoplastic agents devoid of biological effects on CNS can represent a considerable asset for the treatment of non-brain solid tumours [56,57]. However, metal-based chemotherapeutics, headed for almost half a century by cisplatin and congeners, have constantly accomplished a leading role in the treatment of cancer, so that they are still largely used in clinical practice in several oncotherapeutic approaches [1,58]. Nevertheless, serious shortcomings remain to be addressed—mainly linked to the occurrence of severe toxic effects—which represent the main driving force in the search for alternative drugs, equally effective but with improved toxicologic profiles [2]. In this frame, AziRu meets these requirements when chemically linked to ad hoc designed nucleolipid nanosystem, e.g., HoThyRu/DOTAP, blending the benefits of an advanced nanodelivery with those of a Ru-based candidate drug endowed with superior bioactivity. As further confirmation, the HoThyRu/DOTAP nanosystem shows no toxicity on the haematological system. Blood diagnostic profile of treated animals reveals a physiological framework both in terms of blood counts and biomarkers’ activities. Indeed, no significant biochemical, or clinically relevant alterations were observed, either after single administrations or after a one-month dosage regimen, suggesting altogether a good tolerance profile and underscoring the safety of the selected therapeutic protocol. This is in accordance with novel evidence showing cationic liposomes provided with limited toxicity at a low dosage. General investigated parameters such as histopathology, haematology, and clinical chemistry, never showed critical issues at reasonable doses [47,48]. In line, we formerly explored the biocompatibility of DOTAP-based nucleolipid formulation in many human cell lines without observing significant cellular alteration [22,23,40]. The other way around, it is assumed that possible toxic phenomena associated with high doses of cationic liposomes are entirely proportional to their cationic surface charge density, i.e., the more positive cationic liposomes exhibit more critical cytotoxicity [59,60]. Zeta potentials for the characterization of the electrostatic properties of our nanoaggregates—resulting from co-aggregation of the anionic nucleolipid-Ru complex HoThyRu and cationic DOTAP (at a 30:70 ratio)—show that the final liposome composition effectively results in a positive charge. Nevertheless, with respect to bare DOTAP vesicles, the total positive surface charge of HoThyRu/DOTAP is partially neutralized upon addition of the Ru complex, originating a stable nanosystem with no excessive surface charge density [22]. This aspect could be a decisive factor in shaping the toxicological profile of this Ru-based nanosystem, improving its suitability for biomedical applications.

An additional aspect to be addressed concerns the possible activation of inflammatory responses linked to the use of cationic nanosystems [48]. Indeed, experimental evidence reveals cationic liposomes as also capable of leukocytes activation (e.g., neutrophils) by plasma membranes stimulation depending on doses and their intrinsic characteristics, perchance the reason why we found white blood cells profile slightly changed [59,61]. However, even if part of a probably mild inflammatory response, we did not find significant alterations in leukocyte formula, neither after single administrations nor after a one-month dosage regimen. The low liposomal charge ratio can again be critical in mitigating in vivo responses. In view of prospective clinical applications, we will aim at further improving the biocompatibility of these nanosystems, possibly via enhanced selectivity to target cells by fusing specific ligands and/or additional chemical modifications on their surface [50,62].

## 5. Conclusions

By profitably blending amphiphilic nanomaterials as nucleolipids and a Ru(III) complex known as AziRu, we have developed variously decorated anticancer nanosystems which proved to be very effective against human BCC, one of the most widespread human malignancies. Specifically, in the up-to-date panorama of Ru-based candidate drugs we have demonstrated that AziRu, inserted in a nucleolipidic structure and ad hoc nano-delivered by the positively charged lipid DOTAP, can effectively counteract BCC proliferation in vivo while being a well-tolerated agent, which is a crucial property for anticancer drug candidates in preclinical studies to progress in clinical stage. Thus, we have showcased the safety and efficacy of the cationic Ru-based nanosystems termed HoThyRu/DOTAP in a mouse xenograft model of BC. Overall, the herein discussed outcomes validate the use of the HoThyRu/DOTAP nanosystem in an animal model of human BC. Ex vivo investigations are currently ongoing to deepen knowledge on the multi-target action of HoThyRu/DOTAP liposomes on BCC and give additional insights into its mode of action in vivo. Given the need for a next generation metal-based chemotherapeutics to safely fight cancer, these findings are very encouraging in the perspective of a final breakthrough of Ru(III)-based drugs for an upcoming clinic appliance.

## Figures and Tables

**Figure 1 cancers-13-05164-f001:**
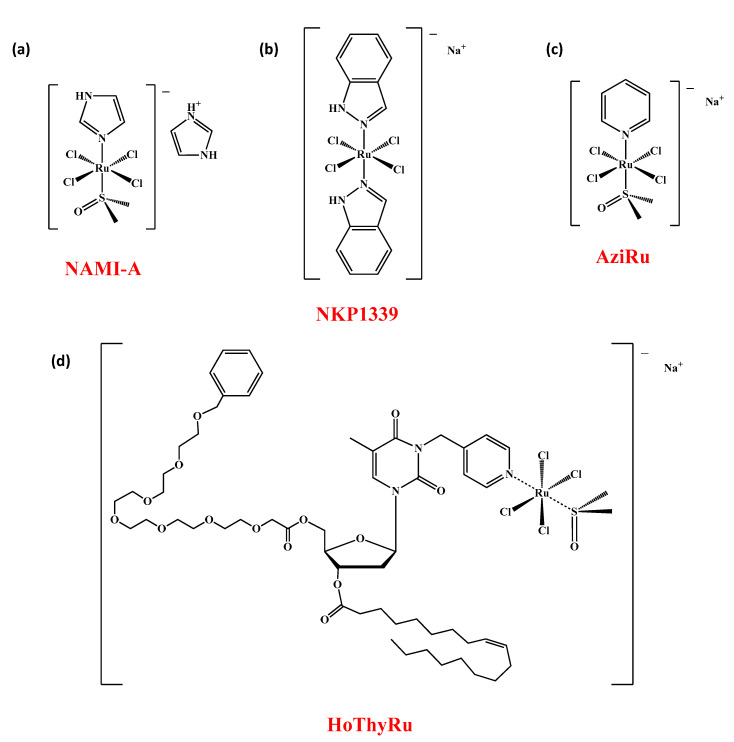
Molecular structures of the low molecular weight ruthenium complexes NAMI-A (**a**), NKP1339 (**b**), AziRu (**c**), and of the HoThyRu nucleolipid-based complex (**d**) incorporating AziRu.

**Figure 2 cancers-13-05164-f002:**
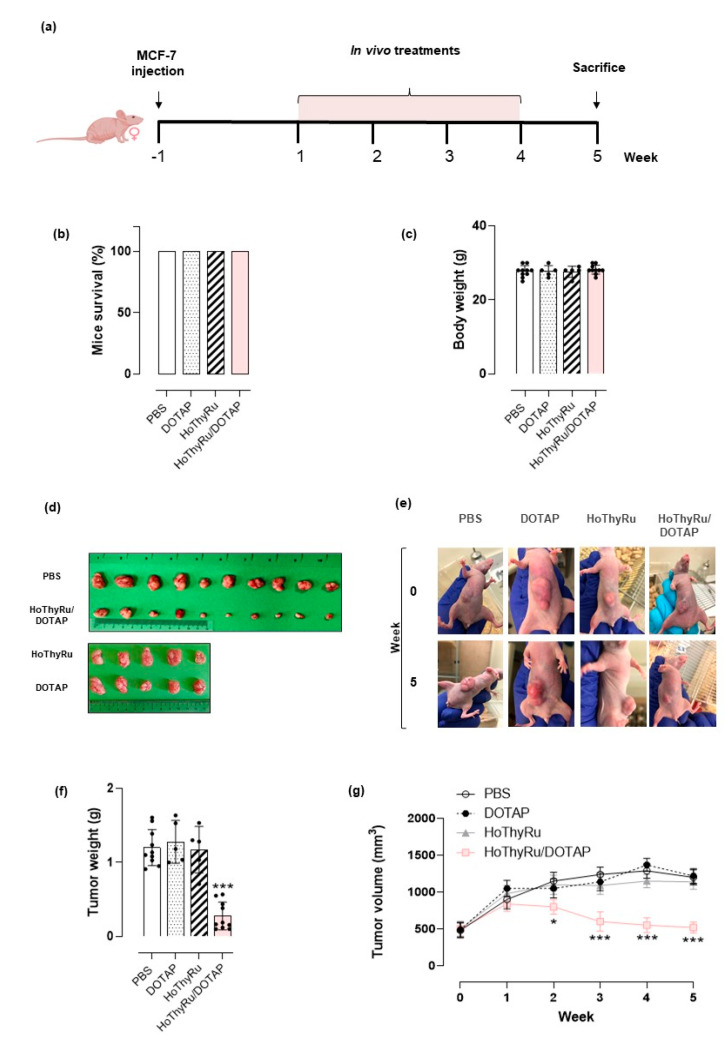
(**a**) Experimental protocol and therapeutic scheme based on intraperitoneal (*i.p.*) administrations of HoThyRu/DOTAP (15 mg/kg), once a week for 28 days. (**b**) Tumour-bearing mice survival and (**c**) body weights at the end of the study (5 weeks from the start of treatments). Control group: animal injected with PBS (Phosphate Buffered Saline pH 7.4, *n* = 10). Treated groups: animal injected with the DOTAP liposome (DOTAP, *n* = 5), the not co-aggregated HoThyRu complex (HoThyRu, *n* = 5), and the HoThyRu/DOTAP nanoformulation (HoThyRu/DOTAP, *n* = 10). (**d**) Explanted tumour masses at the end point of the study from untreated (PBS) and treated (DOTAP, HoThyRu, and HoThyRu/DOTAP) xenotransplanted animal groups. (**e**) Photographs taken at the end of the preclinical trial pertaining to treated (DOTAP, HoThyRu, HoThyRu/DOTAP) and untreated (PBS) xenotransplanted mice showing tumour inhibition by HoThyRu/DOTAP nanoformulation. (**f**) Weight analysis of the explanted tumour masses at the end of the study and (**g**) tumour volumes evaluation over time throughout experiments in mice control group (PBS, *n* = 10 animals) and in mice treated groups (DOTAP, *n* = 5 animals; HoThyRu, *n* = 5 animals; HoThyRu/DOTAP, *n* = 10 animals). Statistical analysis was conducted by one-way ANOVA followed by Bonferroni’s for multiple comparisons. * *p* ≤ 0.05 vs. PBS control group; *** *p* ≤ 0.005 vs. PBS control group.

**Figure 3 cancers-13-05164-f003:**
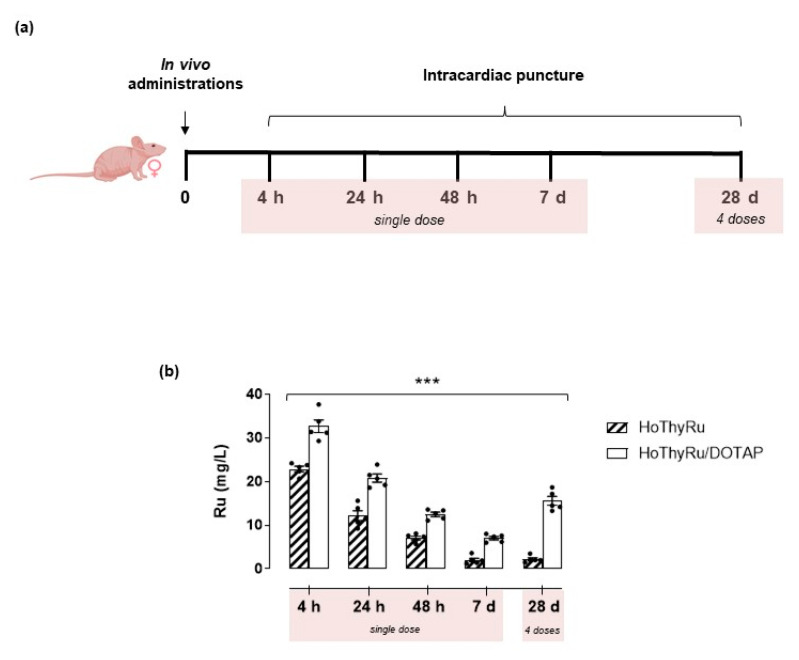
(**a**) Experimental protocol for the preparation of blood samples by intracardiac puncture in nude mice (*n* = 5 animals *per* point) at the indicated times (4, 24, 48 h and 7 days, reported as “single dose”) after a single intraperitoneal administration of HoThyRu (4.5 mg/kg) or HoThyRu/DOTAP (15 mg/kg), or after weekly administrations of HoThyRu (4.5 mg/kg, *i.p.*) and HoThyRu/DOTAP (15 mg/kg, *i.p.*) once a week for 4 weeks (28 d), reported as “4 doses”. (**b**) Evaluation of ruthenium plasmatic levels over time throughout the in vivo study by ICP-MS analysis, as described in the experimental section. Results are plotted in bar graph as mg/L of total ruthenium in mice plasma samples (white bars refer to plasma levels at the indicated times after HoThyRu/DOTAP treatment; ribbed bars refer to plasma levels after HoThyRu treatment). *** *p* ≤ 0.005 vs. the HoThyRu-treated animal group.

**Figure 4 cancers-13-05164-f004:**
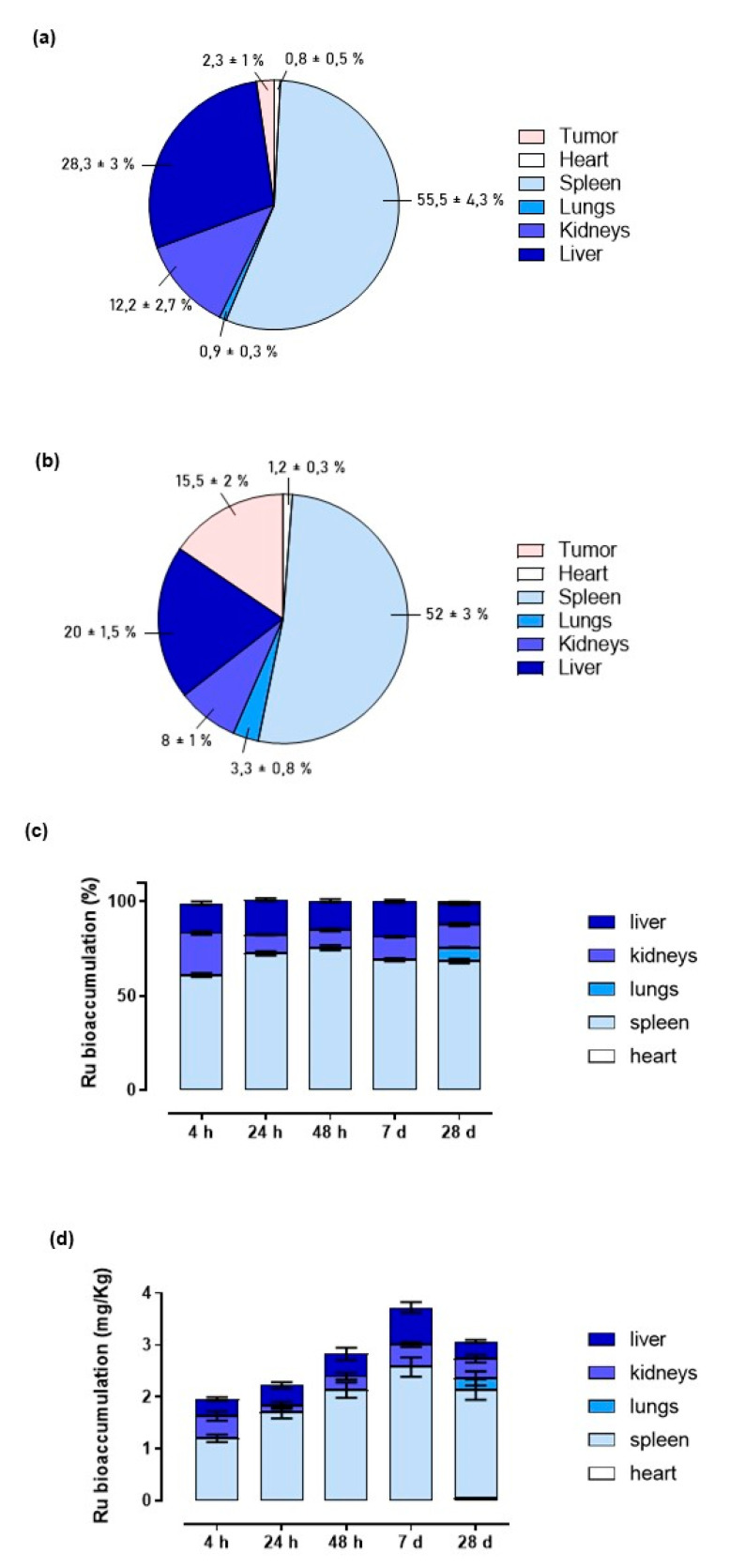
(**a**,**b**) Percentage of ruthenium amounts plotted in pie charts uncovered in the indicated body districts, including tumour lesions, at the endpoint of the preclinical study. After weekly administrations of HoThyRu (**a**) (4.5 mg/kg, *i.p.*, once a week for 4 weeks) and HoThyRu/DOTAP (**b**) (15 mg/kg, *i.p.*, once a week for 4 weeks), the mice were sacrificed, and organs and tissues appropriately collected to analyse the ruthenium content by ICP-MS (*n* = 5 for the HoThyRu-treated group and *n* = 10 for the HoThyRu/DOTAP-treated group). (**c**,**d**) Ruthenium bioaccumulation in mice over time (4, 24, 48 h, and 7 days, as reported in bar graphs) after a single administration of HoThyRu/DOTAP (15 mg/kg, *i.p.*), or after weekly administrations (28 d) of HoThyRu/DOTAP (15 mg/kg, *i.p.*, once a week for 4 weeks) estimated both as (**c**) percentage and (**d**) absolute metal quantity expressed as mg/kg of body weight (*n* = 5 animals *per* time point). Statistical analysis was conducted by one-way ANOVA followed by Bonferroni’s for multiple comparisons.

**Figure 5 cancers-13-05164-f005:**
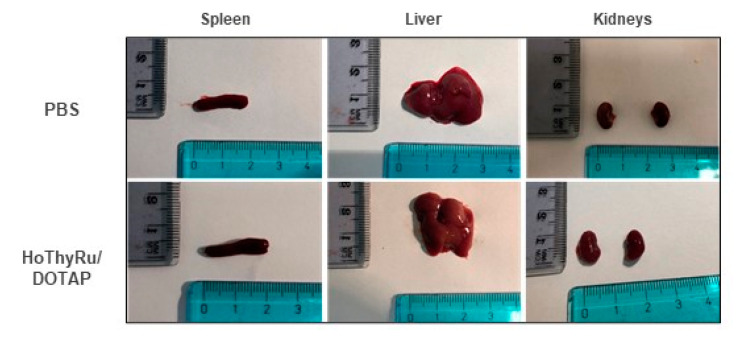
Representative images of spleen, liver and kidneys isolated for autopsy analysis at the endpoint of the preclinical study from control (PBS) and treated mice (HoThyRu/DOTAP, 15 mg/kg, *i.p.*, once a week for 4 weeks).

**Figure 6 cancers-13-05164-f006:**
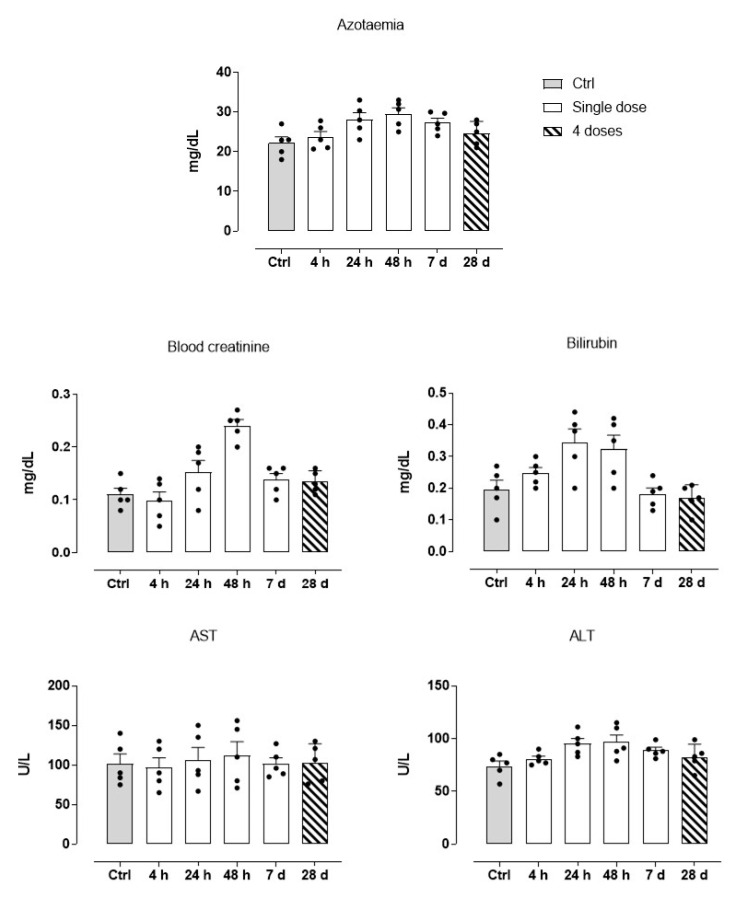
Haematological investigations on blood samples after 4, 24, and 48 h, and 7 days (white bars, “Single dose”) from a single HoThyRu/DOTAP dose (15 mg/kg, *i.p.*), and after weekly administrations (ribbed bars, “4 doses”) of HoThyRu/DOTAP (15 mg/kg, *i.p.*, once a week for 4 weeks), showing the indicated biochemical markers (*n* = 5 animals per time point). Values from control group (non-xenotransplanted mice) were used as reference values and are plotted in graphs as grey bars. Statistical analysis was conducted by one-way ANOVA followed by Bonferroni’s for multiple comparisons.

**Figure 7 cancers-13-05164-f007:**
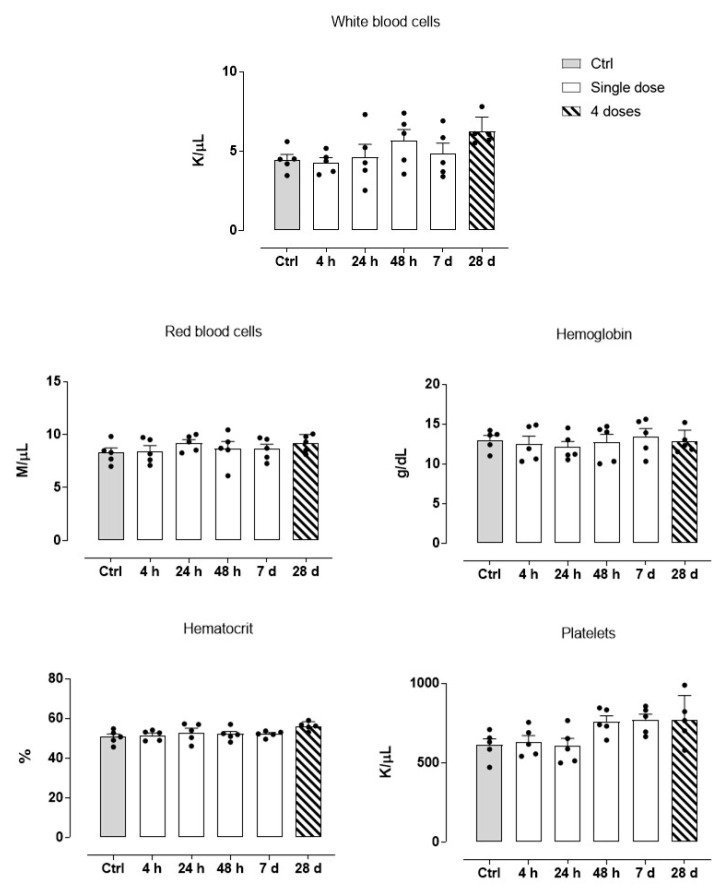
Complete blood count (CBC) test on blood samples taken 4, 24, and 48 h, and 7 days (white bars, “Single dose”) after a single HoThyRu/DOTAP dose (15 mg/kg, *i.p.*), and after weekly administrations (ribbed bars, “4 doses”) of HoThyRu/DOTAP (15 mg/kg, *i.p.*, once a week for 4 weeks), showing the red and white blood cells count, haemoglobin, haematocrit, and total platelets (*n* = 5 animals per time point). Values from control group (non-xenotransplanted mice) were used as reference values and are plotted in graphs as grey bars. Statistical analysis was conducted by one-way ANOVA followed by Bonferroni’s for multiple comparisons.

**Figure 8 cancers-13-05164-f008:**
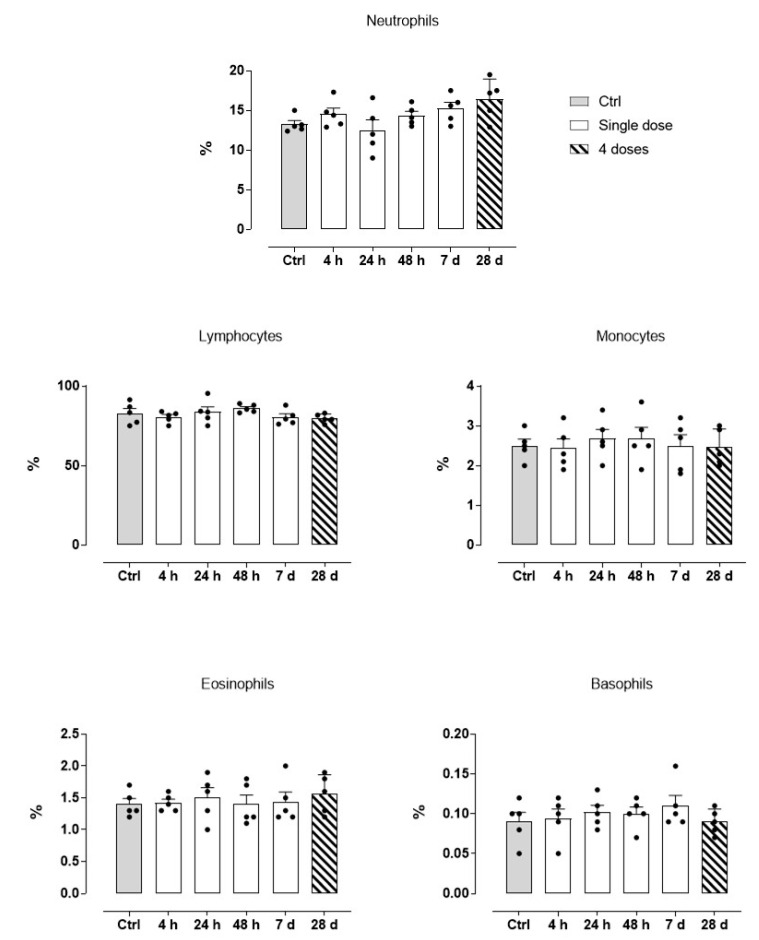
Leukocyte formula determined on blood samples taken 4, 24, and 48 h, and 7 days (white bars, “Single dose”) after a single HoThyRu/DOTAP dose (15 mg/kg, *i.p.*), and after weekly administrations (ribbed bars, “4 doses”) of HoThyRu/DOTAP (15 mg/kg, *i.p.*, once a week for 4 weeks). Values from control group (non-xenotransplanted mice) were used as reference values and are plotted in graphs as grey bars. (*n* = 5 animals per time point). Statistical analysis was conducted by one-way ANOVA followed by Bonferroni’s for multiple comparisons.

## Data Availability

The data presented in this study are available in this article.

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
