# Peer review of "Safety and Efficacy Evaluation In Vivo of a Cationic Nucleolipid Nanosystem for the Nanodelivery of a Ruthenium(III) Complex with Superior Anticancer Bioactivity"

_cancers, 2021, doi:10.3390/cancers13205164_

Round 1

Reviewer 1 Report

The authors correctly addressed the questions raised in the review. The manuscript is now suitable for publication

This manuscript is a resubmission of an earlier submission. The following is a list of the peer review reports and author responses from that submission.

Round 1

Reviewer 1 Report

In the present work, the authors prepared HoThyRu/DOTAP formulations and evaluated their in vivo safety and efficacy in breast cancer tumor-bearing mice. However, the authors already published the efficacy of the current formulations in a breast cancer model in their previous publication (piccolo et al, Scientific reports, 2019, 9, p7006), and hence novelty of the current work is not sufficient. Therefore, I cannot recommend it for publication in  Cancers.

Author Response

Response to Decision letter for cancers-1257091 (article)

“Safety and efficacy evaluation in vivo of a cationic nucleolipid nanosystem for the nanodelivery of a ruthenium(III) complex with superior anticancer bioactivity”

Section - Cancer Therapy

Collection - Cancer Nanomedicine

Point by point author's response to reviewers

The manuscript by Piccolo et al. has considerably benefited according to the reviewers' reports. Their suggestions provided the opportunity to further improve the article. Therefore, we thank the reviewers for their work and valuable advice. In what follows we reply to their comments.

Reviewer #1:

In the present work, the authors prepared HoThyRu/DOTAP formulations and evaluated their in vivo safety and efficacy in breast cancer tumor-bearing mice. However, the authors already published the efficacy of the current formulations in a breast cancer model in their previous publication (Piccolo et al, Scientific reports, 2019, 9, p7006), and hence novelty of the current work is not sufficient. Therefore, I cannot recommend it for publication in Cancers.

Author's Reply to the Review Report (Reviewer 1)

We are a little displeased for Reviewer 1's comment concerning the originality of our manuscript, even though we can somewhat understand his point of view. The work to which he refers (Piccolo et al., Scientific reports, 2019, 9, p7006) is focused almost exclusively on experimental data obtained by preclinical in vitro models deriving from different cellular molecular subtypes of human breast cancer. It is a manuscript in which the main target was to demonstrate autophagy involvement behind the antiproliferative action of a ruthenium (III)  complex incorporated into a cationic nanosystem (HoThyRu/DOTAP, the same nanosystem used for this in vivo study). In Piccolo et al. 2019, in vivo data obtained by a xenograft model of human breast cancer cells are discussed in a small final section within the results. These data were preliminary and just provided to confirm the effectiveness of our formulations in an animal model, thereby allowing to pave the way for new and more comprehensive studies performed in in vivo models. Indeed, as clearly reported in the manuscript cited by Reviewer #1, the focus was to examine the nature of the pathways leading to cell death in human models of breast cancer following ruthenotherapy in vitro. Therefore, based on how the manuscript the referee refers to is structured, the in vivo results were helpful to support some experimental evidence and to provide a first in vivo response. On the contrary, the manuscript currently submitted for publication in Cancers is entirely committed to the preclinical validation of our HoThyRu/DOTAP formulation in an animal model. It is therefore conceived in a totally distinct approach with respect to previous reports. Throughout the development of our original Ru-based nanosystems for cancer treatment, this is the first study based on a comprehensive in vivo research. We agree with the reviewer that some experimental data relating in vivo effectiveness have already been anticipated in an earlier report. However, we are confident that the current manuscript could offer to  Cancers’ readers a significant amount of original data on animal biological responses following in vivo administration of the HoThyRu/DOTAP nanosystem. Additionally, but not least, the experimental approach and therapeutic protocols discussed in the current article are renewed, being based not only on repeated weekly administrations over time but also on single doses in time course experiments to monitor clinical parameters, biomarkers, and other experimental data.

Reviewer 2 Report

The manuscript entitled “Safety and efficacy evaluation in vivo of a cationic nucleolipid 2 nanosystem for the nanodelivery of a ruthenium(III) complex with superior anticancer bioactivity” is a well written and detailed in vivo study of the effect of HoThyRu/DOTAP nanoformulation in mice. I recommend the following minor alterations before publication:

  1. Authors must explain the basis of the first phrase in the Abstract – why tumors are expected to be the first cause of mortality worldwide?
  2. It is not clear in introductory section why the HoThyRu complex was chosen. It would be important to resume the biological action of this complex and the differences between this complex and other Ru(III) complexes such as AziRu
  3. Rewrite the phrase in page 5, line 131-134. The author’s opinions are not relevant in introductory section – in this section state only the facts and major conclusions.
  4. Page 6, section 2.2. – describe the origin of the cell line (where the cell line was obtained)
  5. Page 6, section 2.4. – include a reference for the tripan blue exclusion assay protocol
  6. Page 6, section 2.10. – include a reference or a detailed protocol for the performed biochemical and hematological tests
  7. Prior to the 3.1. section, it would be important to present the characterization of the HoThyRu/DOTAP nanoformulations to confirm that the nanoformulations used in this work are identical to those previously described. Also present in supplementary material the biological in vitro effect of the nanoformulations relative to the free HoThyRu complex and empty DOTAP nanoparticles in MCF-7 cells.
  8. Section 3.1. – justify why this particular therapeutic scheme was chosen.
  9. Section 3.1. – clarify the meaning of the abbreviation “PBS” in the text and in figure caption
  10. Line 329 – detail the exact percentage (average and standard deviation) of Ru in the organs
  11. Figure 4 – include standard deviations and ANOVA.
  12. In figures 3, 6, 7 and 8 captions described the time point where the control sample was obtained and the difference between monodosage and “4 doses” analysis
  13. Page 18, line 502 – do you mean “we found high ruthenium concentrations in plasma even after single administrations”?

Author Response

Response to Decision letter for cancers-1257091 (article)

“Safety and efficacy evaluation in vivo of a cationic nucleolipid nanosystem for the nanodelivery of a ruthenium(III) complex with superior anticancer bioactivity”

Section - Cancer Therapy

Collection - Cancer Nanomedicine

Point by point author's response to reviewers

The manuscript by Piccolo et al. has considerably benefited according to the reviewers' reports. Their suggestions provided the opportunity to further improve the article. Therefore, we thank the reviewers for their work and valuable advice. In what follows we reply to their comments.

Reviewer #2:

The manuscript entitled “Safety and efficacy evaluation in vivo of a cationic nucleolipid nanosystem for the nanodelivery of a ruthenium(III) complex with superior anticancer bioactivity” is a well written and detailed in vivo study of the effect of HoThyRu/DOTAP nanoformulation in mice.

I recommend the following minor alterations before publication:

Author's Reply to the Review Report (Reviewer 2)

  1. Authors must explain the basis of the first phrase in the Abstract – why tumors are expected to be the first cause of mortality worldwide?
  2. It is just a prediction based on epidemiological data by the World Health Organization (WHO). However, to improve accuracy and based on the reviewer # 2 suggestion, we have modified the text in the abstract.

  1. It is not clear in introductory section why the HoThyRu complex was chosen. It would be important to resume the biological action of this complex and the differences between this complex and other Ru(III) complexes such as AziRu.
  2. We thank reviewer #2 for this recommendation to better introduce central features of our most promising formulations occurring throughout preclinical investigations by in vitro models. As described in the introduction section, all our nucleolipid nanosystems, both zwitterionic and cationic (co-aggregated with POPC and DOTAP, respectively), incorporate the bioactive low molecular weight ruthenium complex AziRu and are noticeably much more stable over time than the naked Ru complex, even in a complex situation such as the biological microenvironment. The greater chemical stability, together with extremely favourable cellular uptake kinetics, make our cationic nanosystems much more effective in terms of antiproliferative activity against cancer cells than AziRu and NAMI-A. An extensive scientific literature already reported in the manuscript introduction refers to these concepts and shows what we have uncovered over the last years (references 11-16, 18, 19, 22, 23). However, in line with reviewer # 2, we believe it is appropriate to further improve these basic concepts by adding some sentences substantiating the selection of the HoThyRu/DOTAP nanosystem as the formulation exploited for this in vivo study. Therefore, in the introduction section the manuscript has been improved by incorporating new sentences about some biological features of HoThyRu/DOTAP within the suite of nanoformulations we have developed in recent years (lines 116-124). In any case, many of these concepts are also resumed and deepened in the opening section of the discussion (lines 442-451).

  1. Rewrite the phrase in page 5, line 131-134. The author’s opinions are not relevant in introductory section – in this section state only the facts and major conclusions.
  2. Done. The sentence has been changed.

  1. Page 6, section 2.2. – describe the origin of the cell line (where the cell line was obtained)
  2. Done.

  1. Page 6, section 2.4. – include a reference for the trypan blue exclusion assay protocol
  2. Done. This experimental protocol is discussed extensively in a previous report (Sci Rep. 2017; 7: 45236). Therefore, we have introduced in section 2.4. this reference (ref. 14).

  1. Page 6, section 2.10. – include a reference or a detailed protocol for the performed biochemical and hematological tests
  2. Done. In this section we have better detailed the experimental protocol used for haematological and biochemical tests and we have replaced the reference 34 with a more adequate and recent one.

  1. Prior to the 3.1. section, it would be important to present the characterization of the HoThyRu/DOTAP nanoformulations to confirm that the nanoformulations used in this work are identical to those previously described. Also present in supplementary material the biological in vitro effect of the nanoformulations relative to the free HoThyRu complex and empty DOTAP nanoparticles in MCF-7 cells.
  2. Based on the reviewer's suggestions, we have included a DLS control of the HoThyRu/DOTAP nanosystem (at a 30:70 ratio) as supplementary material (Fig. S1), introducing it before section 3.1 within the materials and methods (section 2.1. - Preparation of HoThyRu/DOTAP formulation). As well, the experimental data required by the reviewer concerning in vitro biological effects on MCF-7 cells prior to in vivo administrations have been included as supplementary material (Fig. S2), introduced in both section 2.2. (Cell cultures) and in section 3.1 of results.

  1. Section 3.1. – justify why this particular therapeutic scheme was chosen.
  2. Starting from an original research proposal submitted to the Italian Higher Institute of Health, the therapeutic protocol used for this study is the one that has been authorized for animal testing. Further details can be found in section 2.3. (Animals and experimental design)

  1. Section 3.1. – clarify the meaning of the abbreviation “PBS” in the text and in figure caption
  2. Done. As highlighted in the text, minor changes have been made to clarify the meaning of PBS in both the text and Fig. 2 caption.

  1. Line 329 – detail the exact percentage (average and standard deviation) of Ru in the organs
  2. Done.

  1. Figure 4 – include standard deviations and ANOVA.
  2. Done. A new and improved Fig. 4 was included in the revised form of the manuscript.

  1. In figures 3, 6, 7 and 8 captions, describe the time point where the control sample was obtained and the difference between monodosage and “4 doses” analysis
  2. Done. In the legends of the figures in question we have now more clearly indicated the difference between the animal experimental groups (“Single dose” and “4 doses”). Regarding the control samples, these analyses were performed on biological samples by the animal control group (non-xenotransplanted mice), and therefore we have not shown the timing. As also described in the experimental section, these are simply biological samples carried out randomly from the group of control animals.

  1. Page 18, line 502 – do you mean “we found high ruthenium concentrations in plasma even after single administrations”?
  2. Yes. Thanks for this clarification.

Reviewer 3 Report

Comments

The authors have developed cationic Ru(III)-based nucleolipid formulation, named HoThyRu/DOTAP and have shown anti-tumor effect and safety using breast tumor-bearing mouse model.

Major comments:

1: Information for the physical and chemical properties of this nanoparticles, HoThyRu/DOTAP, including drug loading efficiency and release from the liposomes is missing.

2: In vitro evaluation of anti-cancer cell effect by HoThyRu/DOTAP and HoThyRu is missing.

4: In vivo study, for therapy experiment using xenotransplanted mice, there are only two treatment groups, PBS and HoThyRu/DOTAP. As proper control, tumor-bearing mice shall be treated with non-nanoparticle form of therapeutics, HoThyRu and drug-free liposomes (DOTAP).

5: Drug delivery advantage to tumor using HoThyRu/DOTAP shall be compared with that using HoThyRu.

5: In vitro and or in vivo evaluation of the therapeutic effect on apoptotic pathway, which is discussed as the major target of the therapeutic, is missing.

Author Response

Response to Decision letter for cancers-1257091 (article)

“Safety and efficacy evaluation in vivo of a cationic nucleolipid nanosystem for the nanodelivery of a ruthenium(III) complex with superior anticancer bioactivity”

Section - Cancer Therapy

Collection - Cancer Nanomedicine

Point by point author's response to reviewers

The manuscript by Piccolo et al. has considerably benefited according to the reviewers' reports. Their suggestions provided the opportunity to further improve the article. Therefore, we thank the reviewers for their work and valuable advice. In what follows we reply to their comments.

Reviewer #3:

The authors have developed cationic Ru(III)-based nucleolipid formulation, named HoThyRu/DOTAP and have shown anti-tumor effect and safety using breast tumor-bearing mouse model.

Author's Reply to the Review Report (Reviewer 3)

  • Information for the physical and chemical properties of this nanoparticles, HoThyRu/DOTAP, including drug loading efficiency and release from the liposomes is missing.
  • We thank reviewer# 3 for his work and valuable suggestions, which allowed to further improve this article. Based on the reviewer's recommendations, we have now included a DLS control as physico-chemical characterization of the HoThyRu/DOTAP nanosystem (at a 30:70 ratio) in supplementary material (Fig. S1) launched in Materials and Methods (section 2.1. - Preparation of HoThyRu/DOTAP formulation). Nevertheless, a large, cited literature referring to former reports about the physico-chemical features of our nanosystems - including the HoThyRu/DOTAP cationic nanoformulation used for this study - is already enclosed in the manuscript. Reference 22 (Biomacromolecules, 2013; 14(8): 2549-2560) describes exhaustively the chemical, physical and biological characterization of nucleolipid nanosystems co-aggregated with the positively charged lipid DOTAP. Similarly, reviews 11, 12 and 16 are plenty of data on structural and biological properties of our cationic nanosystems.

  • In vitro evaluation of anti-cancer cell effect by HoThyRu/DOTAP and HoThyRu is missing.
  • Based on the reviewer's suggestions, we have also integrated as supplementary material (Fig. S2) the experimental data concerning in vitro biological effects on MCF-7 cells prior to in vivo administrations, introducing them within the materials and methods (section 2.2. – Cell cultures) and in section 3.1 of results.

  • In vivo study, for therapy experiment using xenotransplanted mice, there are only two treatment groups, PBS and HoThyRu/DOTAP. As proper control, tumor-bearing mice shall be treated with non-nanoparticle form of therapeutics, HoThyRu and drug-free liposomes (DOTAP).
  • We agree with the reviewer on this point. In accordance with the guidelines and policies of the European Communities Council on animal experimentation, in the original experimental proposal submitted to the Italian Higher Institute of Health to get the required authorizations for in vivo studies, we also contemplated various controls for preclinical investigations, including mice treated with the bare DOTAP liposomes and/or the nucleolipid complex HoThyRu. Different types of experimental protocols have also been submitted for final approval. However, to reduce and limit the number of animals, the project was revised, and we took approval to use a smaller number of mice as control groups. Therefore we performed experiments only on an additional control group represented by mice treated in vivo with the bare DOTAP liposomes. Having just available this kind of data for some preliminary in vivo investigations and following reviewer suggestions, we have included in the new version of the manuscript a fifth experimental group consisting of xenotransplanted animals treated with DOTAP liposomes. For this reason, Materials and Methods have been specially modified (section 2.1, 2.3 and 2.5), as well as results (section 3.1), where we have introduced a short sentence to explain the presence of this additional control. Also Fig. 2 has obviously been replaced by a new figure 2 which contemplates changes in points (b), (c), (f) and (g) (DOTAP liposome as additional control group).              To fully reply to the reviewer’s inquiries, in the original project submitted to the Italian Ministry of Health, all data emerged during the preclinical investigations by cellular models were taken into consideration to establish an in vivo procedure. In this context, the most bioactive formulation was the co-aggregated one by the lipid DOTAP (HoThyRu/DOTAP) compared to the nucleolipid complex HoThyRu and the low molecular weight ruthenium complex AziRu, which were found to be devoid of significant bioactivity in both tumor cells (including MCF-7) and control cells. These results are extensively covered in Biomacromolecules, 2013; 14 (8): 2549-2560 (ref. 22 in the current manuscript).                                                                                            Finally, the achievement of this in vivo study herein submitted has required considerable financial resources considering the animal models setting up starting from immunosuppressed nude mice (50 animals used in total throughout the study).

  • Drug delivery advantage to tumor using HoThyRu/DOTAP shall be compared with that using HoThyRu.
  • As also discussed in the previous point, the biological characterization of our nanosystems co-aggregated with DOTAP or not, their interaction, uptake and cellular trafficking have been extensively covered in the experimentation of these nanosystems in cellular models. As reported in the manuscript, both in introduction and discussion, these data are available in the numerous references that have been inserted about these topics (in particular, reference 22: Biomacromolecules, 2013; 14(8): 2549-2560). Considering the reviewer's request at point 2, we have included in Figure S2 data relating to the comparison between the HoThyRu nucleolipid complex and the final HoThyRu/DOTAP nanosystem, which demonstrate the superior efficacy of the latter in inhibiting growth and proliferation of MCF-7 cells.

  • In vitro and or in vivo evaluation of the therapeutic effect on apoptotic pathway, which is discussed as the major target of the therapeutic, is missing.
  • The study of death pathways activated in cellular models by these ruthenium-based nanosystems, including the apoptotic ones, has been the subject of previous papers on this topic (ref. 14, 15, 16, 40). Different molecular subtypes of human breast cancer have been used for this purpose, including TNBC models such as MDA-MB-231 cells, where we have demonstrated induction of both mitochondrial apoptosis and autophagy (ref. 15, Sci Rep. 2019; 9(1): 7006.). In the manuscript submitted to Cancers constant reference is made to these former studies. Moreover, the evaluation in preclinical models in vivo and ex vivo of the molecular effects of these anticancer agents in tumor cells will be the subject of a new paper entirely committed to the study and characterization of these molecular pathways.

Round 2

Reviewer 2 Report

  • In the submitted document, contrary to the author’s answer, the text relative to the question “why tumors are expected to be the first cause of mortality worldwide?” was not altered. The affirmation that "tumors are expected to be the first cause of mortality soon" is speculative with no fundaments. Remove this sentence from the abstract.
  • When referring the phrases “Among these, the cationic AziRu-based nucleolipid formulation named HoThy-Ru/DOTAP has been specifically designated for this study. Indeed, in the final nanoformulation the positively charged lipid DOTAP proved high performance in stabilizing and delivering the HoThyRu complex (the molecular structure of the HoThyRu nucleolipid compound incorporating the AziRu complex is shown in Fig. 1).” Refer the study where this formulation was analysed. As it is, it is not clear the mechanism of action of HoThyRu, only that the compound per se have antiproliferative activity against BC cells.
  • It is not understandable why authors use the reference 16, which is a review, after the phrase “In turn, HoThyRu has demonstrated remarkable antiproliferative bioactivity in solid tumours such as BC, associated with ruthenium IC50 values in the low micromolar range (e.g., 12 μM in both the human endocrine-responsive epithelial-like type breast adenocarcinoma MCF-7 cells and the TNBC MDA-MB-231 cells)”
  • The city and country of ATCC is missing
  • Contrary to the author’s answer, the text was not highlighted to clarify the meaning of PBS

Reviewer 3 Report

1: Information for the physical and chemical properties of this nanoparticles, HoThyRu/DOTAP, including drug loading efficiency and release from the liposomes is missing.

Reviewer’s comment: The author responded to this comment and added enough data and revised text.

2: In vitro evaluation of anti-cancer cell effect by HoThyRu/DOTAP and HoThyRu on cancer cell line(s) is missing.

Reviewer’s comment: The author responded to this comment and performed in vitro study shown in the supplementary material (Fig. S2) and revised the text.

4: In vivo study, for therapy experiment using xenotransplanted mice, there are only two treatment groups, PBS and HoThyRu/DOTAP treated groups. As proper control, tumor-bearing mice shall be treated with non-nanoparticle form of therapeutics, HoThyRu and drug-free liposomes (DOTAP).

The authors explained difficulty to perform in vivo study as follows:

Different types of experimental protocols have also been submitted for final approval. However, to reduce and limit the number of animals, the project was revised, and we took approval to use a smaller number of mice as control groups. Therefore we performed experiments only on an additional control group represented by mice treated in vivo with the bare DOTAP liposomes.”

Reviewer’s comment: In the revised manuscript, the number of mice for the new control group treated with DOTAP liposomes is still 10, which is same as that for the other groups and the number has not been reduced. If the number of mice for control groups had to be reduced from 10, the authors can use these unused mice for additional and necessary control group of mice treated with non-nanoparticle form of therapeutics, HoThyRu. Also, statistical power analysis to determine necessary number of mice for the study is not shown. Because of the result shown in Fig2 (g) with small error bar, 10 mice may not be necessary.

Authors also mentioned about cost for the in vivo study, but this type of explanation shall not be taken into account for scientific evaluation of the paper.

 5: Drug delivery advantage to tumor using HoThyRu/DOTAP shall be compared with that using HoThyRu.

The authors explained as: the biological characterization of our nanosystems co-aggregated with DOTAP or not, their interaction, uptake and cellular trafficking have been extensively covered in the experimentation of these nanosystems in cellular models.

Reviewer’s comment: Drug delivery of nanoparticle-based therapeutics and free encapsulated therapeutics in vivo shall be different from that in in vitro. This will be one of the major interests in this paper. Ruthenium bioaccumulation in mice having BBC xenograft (Fig 4) shall be measured and compared using two formulations.  

5: In vitro and or in vivo evaluation of the therapeutic effect on apoptotic pathway, which is discussed as the major target of the therapeutic, is missing.

The authors explained as: the evaluation in preclinical models in vivo and ex vivo of the molecular effects of these anticancer agents in tumor cells will be the subject of a new paper entirely committed to the study and characterization of these molecular pathways.

Reviewer’s comment: I don’t think it is a reasonable explanation for not having the data in the current paper.